# Desferrioxamine B-Mediated Pre-Clinical In Vivo Imaging of Infection by the Mold Fungus *Aspergillus fumigatus*

**DOI:** 10.3390/jof7090734

**Published:** 2021-09-08

**Authors:** Matthias Misslinger, Milos Petrik, Joachim Pfister, Isabella Hubmann, Katerina Bendova, Clemens Decristoforo, Hubertus Haas

**Affiliations:** 1Institute of Molecular Biology, Biocenter, Medical University of Innsbruck, 6020 Innsbruck, Austria; matthias.misslinger@i-med.ac.at; 2Institute of Molecular and Translational Medicine, Faculty of Medicine and Dentistry, Palacky University, 77900 Olomouc, Czech Republic; milos.petrik@upol.cz (M.P.); katerina.bendova01@upol.cz (K.B.); 3Department of Nuclear Medicine, Medical University of Innsbruck, 6020 Innsbruck, Austria; joachim.pfister@i-med.ac.at (J.P.); isabella_hubmann@gmx.at (I.H.)

**Keywords:** *Aspergillus fumigatus*, infection, invasive pulmonary aspergillosis, imaging, siderophore, desferrioxamine B, pH, gallium-68, positron emission tomography, PET

## Abstract

Fungal infections are a serious threat, especially for immunocompromised patients. Early and reliable diagnosis is crucial to treat such infections. The bacterially produced siderophore desferrioxamine B (DFO-B) is utilized by a variety of microorganisms for iron acquisition, while mammalian cells lack the uptake of DFO-B chelates. DFO-B is clinically approved for a variety of long-term chelation therapies. Recently, DFO-B-complexed gallium-68 ([^68^Ga]Ga-DFO-B) was shown to enable molecular imaging of bacterial infections by positron emission tomography (PET). Here, we demonstrate that [^68^Ga]Ga-DFO-B can also be used for the preclinical molecular imaging of pulmonary infection caused by the fungal pathogen *Aspergillus fumigatus* in a rat aspergillosis model. Moreover, by combining in vitro uptake studies and the chemical modification of DFO-B, we show that the cellular transport efficacy of ferrioxamine-type siderophores is impacted by the charge of the molecule and, consequently, the environmental pH. The chemical derivatization has potential implications for its diagnostic use and characterizes transport features of ferrioxamine-type siderophores.

## 1. Introduction

Invasive fungal infections, such as aspergillosis, bear a high risk for fatal outcomes. These infections are difficult to diagnose and treat. Due to insufficient diagnoses, immunocompromised patients are often subject to empirical therapy with azoles, the main antifungal therapy [1,2]. Accurate in-time detection of fungal infections could minimize the need for prophylaxis and would optimize therapy. The gold standard for the diagnosis of fungal infections, such as invasive aspergillosis, is to culture the fungus from biological samples. This technique comes with advantageous yield of the specific pathogen, which allows testing for susceptibility to drugs used for treatment [3]. However, cultures from the blood samples of invasive aspergillosis patients are usually negative [4]. In addition, biomarkers, such as galactomannan, (1–>3)-beta-D-glucan and DNA, are often not detected in blood samples in the early stages of invasive *Aspergillus* infections. For the early detection of *Aspergillus* infections, tissue biopsy or bronchoalveolar lavage fluid is required. These procedures are laborious, time-consuming and often lack sensitivity [5]. Hence, time-saving diagnosis methods are required, as the speed fungal infection detection often determines the outcome of a fungal infection [6].

Siderophores are low-molecular mass iron chelators, which are secreted in response to iron limitation by most bacteria and fungi to support iron acquisition. Upon the chelation of iron from the fungal environment, iron-loaded siderophores are taken up by siderophore-specific transporters. During an infection, this mechanism allows the pathogen to steal iron from the host. Hence, siderophore uptake mechanisms are activated by the pathogen during infection [7,8].

The major cause of aspergillosis is *Aspergillus fumigatus* [9]. This mold produces four types of siderophores: intracellular ferricrocin (FC) and hydroxyferricrocin, and secreted fusarinine C and its acetylated form triacetylfusarine C (TAFC). It has been demonstrated that siderophore biosynthesis is essential for the virulence of *A. fumigatus* in a murine aspergillosis model, demonstrating the activity of siderophore-mediated iron acquisition during infection by this mold [8,10]. Similar to most microorganisms, *A. fumigatus* is not only able to utilize self-produced siderophores, but also siderophores produced by other species (xenosiderophores), such as ferrioxamine B and ferrioxamine E [11,12]. Ferrioxamines were originally identified as iron-bound chelates [13]. Later, the iron-free versions of ferrioxamines were named desferrioxamines, such as desferrioxamine B (also termed deferoxamine, DFO-B) and desferrioxamine E (DFO-E), which are hydroxamate-type siderophores produced by bacteria such as streptomycetes or *Erwinia* spp. [14,15]. The chemical structures of TAFC, DFO-B and DFO-E are displayed in Figure 1. In fungal species, siderophore-metal complexes are taken up by members of the so-called “Siderophore Iron Transporter (SIT)” family, which is a subgroup of the major facilitator protein superfamily. SITs are found in virtually all fungal species but their presence is confined to the fungal kingdom, as mammals lack orthologs [8].

Siderophores are best known for forming complexes with Fe^3+^ as needed for iron acquisition. Due to their similar properties, these chelators can also complex Ga^3+^, an isosteric diamagnetic substitute for Fe^3+^ [16,17]. ^68^Ga is a positron emitter which can be monitored by positron emission tomography (PET). It is widely available using a ^68^Ge/^68^Ga-generator and, with a half-life of 68 min, ^68^Ga results in a low radiation burden for the patient. TAFC- and DFO-E-complexed ^68^Ga were previously shown to allow the molecular imaging of pulmonary *A. fumigatus* infections in a rat model [11]. However, TAFC and DFO-E are not yet approved for clinical applications.

DFO-B (trade name Desferal^®^) is an approved drug that has been used in clinics for decades for metal chelation therapy in beta-thalassemia, sideroblastic anaemia, auto-immune haemolytic anemia and other iron or aluminum overload pathologies [18,19,20]. Recently, DFO-B-complexed ^68^Ga ([^68^Ga]Ga-DFO-B) was shown to be suitable for the molecular imaging of bacterial infections in animal infection models [21]. *A. fumigatus* was shown to utilize DFO-B, however, with significantly lower efficacy compared to the TAFC and DFO-E metal complexes in vitro [12]. Therefore, to date, approaches for the siderophore-mediated imaging of fungal infections, such as aspergillosis, have focused on TAFC and DFO-E. Recently, DFO-B was shown to have in vitro characteristics and pharmacokinetics in both mice and rats, qualifying it as an excellent radiotracer [21]. DFO-B can be easily radiolabeled with ^68^Ga, and [^68^Ga]Ga-DFO-B exhibits low protein binding, high stability in human serum and PBS, rapid renal excretion, and minimal retention in the blood and other organs in healthy mice. Moreover, [^68^Ga]Ga-DFO-B was proven to be suitable for the in vivo PET imaging of bacterial infections in mice and rats [21]. DFO-E is an uncharged cyclic desferrioxamine, while DFO-B is a linear derivative with a terminal amine-group that is positively charged at an acidic pH (Figure 1). Thus, we aimed to evaluate (i) [^68^Ga]Ga-DFO-B for the in vivo imaging of aspergillosis in a rat infection model, (ii) the in vivo uptake of Ga-DFO-B versus Ga-DFO-E and (iii) the possibility of a pH-dependent charge affecting the uptake of DFO-B and modified DFO-B derivatives in comparison to the uncharged DFO-E.

## 2. Materials and Methods

### 2.1. Animal Experiments

A previously described rat model of pulmonary aspergillosis was used for the in vivo experiments [22]. The introduction of *A. fumigatus* spores into animals, injection of radiolabeled siderophores and small animal imaging were carried out under 2% isoflurane anesthesia (FORANE, Abbott Laboratories, Abbott Park, IL, USA) to minimize animal suffering and prevent animal motion. Infection in the lung of immunodeficient rats was established by intratracheal inoculation of 100 μL of *A. fumigatus* spores (10^9^ CFU/mL *A. fumigatus* ATCC 46645; American Type Culture Collection) using the TELE PACK VET X LED system equipped with a flexible endoscope (https://www.karlstorz.com/cps/rde/xbcr/karlstorz_assets/ASSETS/3386919.pdf, accessed on 20 August 2021; Karl Storz GmbH & Co. KG, Tuttlingen, Germany). Depending on the development of the infection, experimental animals underwent an ex vivo biodistribution study or PET/CT imaging, typically 2–4 days after inoculation.

### 2.2. Ex Vivo Biodistribution

*A. fumigatus*-infected rats were retro-orbitally (r.o.) injected with [^67^Ga]- and [^68^Ga]Ga-DFO-B and DFO-E. One infected rat (rat 1) was injected with [^68^Ga]Ga-DFO-B and [^67^Ga]Ga-DFO-E, while the second rat (rat 2) was infected with [^67^Ga]Ga-DFO-B and [^68^Ga]Ga-DFO-E (~0.5 MBq and ~0.5 µg of each DFO-B and DFO-E per rat). Rats were sacrificed by exsanguination 45 min post-injection (p.i.). Organs and tissues of interest (blood, spleen, pancreas, kidneys, liver, heart, muscle and lung) were removed and weighed. The amount of radioactivity in the samples was measured in a γ-counter (2480 Wizard-2; Perkin Elmer, Waltham, MA, USA). First, a short-lived ^68^Ga was measured immediately after organ removal in the 511 ± 100 keV energy window, and ^67^Ga was measured after 24 h in the 185 ± 100 keV energy window. The results are expressed as percentage of injected dose per gram of organ (%ID/g).

### 2.3. Animal Imaging

µPET/CT images were acquired with an Albira PET/SPECT/CT small animal imaging system (Bruker Biospin Corporation, Woodbridge, CT, USA). Anaesthetized *A. fumigatus* infected rats were r.o. injected with [^68^Ga]Ga-DFO-B in a dose of ~5 MBq corresponding to ~4 μg of DFO-B per animal. Static PET/CT imaging was performed 45 min p.i. A 10 min PET scan (axial FOV 148 mm) was followed by a 30 min CT scan (axial FOV~160 mm, 45 kVp, 400 μA, at 400 projections). Scans were reconstructed with the Albira software (Bruker Biospin Corporation, Woodbridge, CT, USA) using the maximum likelihood expectation maximization (MLEM) and filtered back-projection (FBP) algorithms. After reconstruction, acquired data were viewed, analyzed and processed using the appropriate software (PMOD software, PMOD Technologies Ltd., Zurich, Switzerland and VolView software, Kitware, Clifton Park, NY, USA).

### 2.4. Synthesis of the Derivatives

#### 2.4.1. Analytical RP-HPLC

Reversed-phase (RP) high-performance liquid chromatography (HPLC) analysis was carried out using an UltiMate 3000 RS UHPLC pump, UltiMate 3000 autosampler, UltiMate 3000 Variable Wavelength Detector (UV detection at λ = 220 nm (Dionex, Germering, Germany)), and radio-detector (Gabi Star, Raytest; Straubenhardt, Germany). We used Jupiter 5 μm C18 300 Å 150 × 4.6 mm (Phenomenex Ltd. Aschaffenburg, Germany) as the column, with acetonitrile (ACN)/H_2_O/0.1% trifluoroacetic acid (TFA) as the mobile phase. Gradient A: 0.0–3.0 min 10% ACN, 3.0–16.0 min 10–60% ACN, 16.0–18.0 min 60% ACN, 18.0–18.1 min 60–10% ACN, 18.1–22.0 min 10% ACN; flow rate of 1 mL/min.

#### 2.4.2. Preparative RP-HPLC

Sample purification via RP-HPLC was carried out on a Gilson 322 Pump with a Gilson UV/VIS-155 detector (UV detection at λ = 220 nm) using a PrepFC™ automatic fraction collector (Gilson, Middleton, WI, USA), Eurosil Bioselect Vertex Plus 30 × 8 mm 5 μm C18A 300 Å pre-column and Eurosil Bioselect Vertex Plus 300 × 8 mm 5 μm C18A 300 Å column (Knauer, Berlin, Germany) and ACN/H_2_O/0.1% TFA as the mobile phase. Gradient 1: 0.0–1.0 min 0% ACN, 1.0–35.0 min 0–50% ACN, 35.0–36.0 min 50% ACN, 36.0–36.1 min 50–0% ACN, 36.1–43.0 min 0% ACN; flow rate of 2 mL/min.

#### 2.4.3. Fe-DFO-B

DFO-B was purchased as commercially available Desferal^®^ (Novartis Pharma GmbH, Vienna, Austria). Iron-containing Fe-DFO-B was obtained by addition of 10-fold molar excess of FeCl_3_ in aqueous solution to DFO with subsequent purification by preparative HPLC.

#### 2.4.4. Fe-DFO-B^ac^ and Fe-DFO^suc^

First, 30 mg (54 µmol) of Fe-DFO-B was dissolved in 500 µL of MeOH and mixed with either 200 µL (2.12 mmol) acetic anhydride or 28 mg (280 µmol) succinic anhydride. Subsequently, the pH level was adjusted to 8 with N,N-Diisopropylethylamine (DIPEA). The mixtures were shaken for 10 min, and the end of the reaction was confirmed by analytical RP-HPLC. The solutions were immediately purified by preparative RP-HPLC. After removing the solvent in a vacuum, 50% of each product was incubated with 1 mL of Na_2_EDTA (100 µM) and stirred for 2 h to remove iron from the complexes. The compounds were again purified by preparative RP-HPLC and lyophilized, resulting in DFO-B^ac^ and DFO-B^suc^.

#### 2.4.5. Analytical Data

Fe-DFO-B^ac^ 8.15 mg [12.4 µmol, 46%], RP-HPLC gradient A: t_R_ = 8.7 min; mass: *m*/*z* [M + H] = 656.28, [C_27_H_47_FeN_6_O_9_; calculated mass: 655.27]; Fe-DFO^suc^ 23.48 mg [32.9 µmol, 54%]; RP-HPLC gradient A: t_R_ = 9.7 min; Mass: *m*/*z* [M + H] = 714.28, [C_29_H_49_FeN_6_O_11_; calculated mass: 713.28].

### 2.5. Radiolabelling

Fractionated elution of ^68^Ge/^68^Ga-generator (IGG100. Eckert and Ziegler Isotope Products, Berlin, Germany; nominal activity of 1850 MBq) with 0.1 M hydrochloric acid (HCL, Rotem Industries, Arva, Israel) was used to obtain ^68^GaCl_3_ (~250 MBq) in 1.5 mL eluate. For labelling, 10 μg (5–8 nmol) of siderophore were mixed with 200 μL gallium eluate (~15–30 MBq), and the pH was adjusted to 4.5 by adding 20 μL of the sodium acetate solution (1.14 M) per 100 μL eluate. The mixture was left to react for 10 min at room temperature and finally analyzed by radio-TLC (Scan-RAM™, LabLogistic, Sheffield, UK) and radio-RP-HPLC, as described by the authros of [21]. For ^67^Ga-labelling, 20 μg of DFO-B or DFO-E was mixed each with 100 μL of ^67^Ga-citrate (KC Solid, Rokycany, Czech Republic) and incubated for 20 min at 90 °C.

### 2.6. Short-Term Siderophore Uptake Analysis

Short-term uptake of siderophore-complexed ^68^Ga was analyzed as described previously [23,24]. *A. fumigatus* strain Afs77 [25] was grown for 18 h at 37 °C in 100 mL *Aspergillus* minimal medium (AMM) according to Pontecorvo et. al. [26] with 1% (*w*/*v*) glucose as the carbon source and 20 mM glutamine as the nitrogen source, using iron-free trace elements to induce iron limiting for the transcriptional activation of siderophore uptake [27]. To generate iron-replete mycelia for the transcriptional downregulation of siderophore uptake, the growth medium contained an additional 0.03 mM FeSO_4_. For the analysis of the pH-dependent siderophore uptake, the pH of the fungal culture medium was adjusted with HCl (1 M) and NaOH (1 M) immediately before the uptake analyses. Uptake assays were performed with 180 µL of liquid *A. fumigatus* culture added to a prewetted 96-well MultiScreen Filter Plates HTS (1 μm glass fiber filter, Merck Millipore, Darmstadt, Germany). Then, 50 µL of radioactive labelled siderophores (~100 nM) were added to the culture and incubated for 45 min at 37 °C. After washing two times with ice-cold TRIS buffer, the dried filters were measured in a γ-counter.

### 2.7. Analysis of Siderophore Utilization Via Growth Promotion

Growth assays were performed on solid *Aspergillus* minimal medium (AMM), as described above, solidified with 1.8% (*w*/*v*) agar. The pH was adjusted with NaOH, and the medium was supplemented with the respective siderophore. The agar plates were point-inoculated with 10^4^ spores per dot of ∆*sidA*∆*ftrA* [10] and incubated at 37 °C for 48 h.

## 3. Results

### 3.1. [^68^Ga]Ga-DFO-B Allows In Vivo PET Imaging of Pulmonary Aspergillosis in a Rat Model

To test the potential of [^68^Ga]Ga-DFO-B for the imaging of a pulmonary *A. fumigatus* infection in a rat aspergillosis model, DFO-B was radiolabeled with ^68^Ga with the molar activity exceeding 3 GBq/μmol and radiochemical purity >  95% confirmed by both RP-HPLC and ITLC-SG methods in accordance with our previous finding [21]. After the r.o. injection of [^68^Ga]Ga-DFO-B in infected rats, PET/CT imaging allowed a clear depiction of the infected area in the lungs, while unbound [^68^Ga]Ga-DFO-B was rapidly eliminated via the renal system (Figure 2). The quality of imaging was comparable to images using [^68^Ga]Ga-DFO-E in previous studies [11].

### 3.2. In Vivo Uptake of [^67/68^Ga]Ga-DFO-B Is Comparable to [^67/68^Ga]Ga-DFO-E

To further compare [^68^Ga]Ga-DFO-B and [^68^Ga]Ga-DFO-E for targeting pulmonary aspergillosis, two immunocompromised rats were infected with *A. fumigatus* and injected with a combination of DFO-B and DFO-E radiolabeled with short-lived (^68^Ga; t_1/2_ = 68 min) or longer-lived (^67^Ga; t_1/2_ = 78.3 h) radionuclides, respectively, allowing the discrimination of the two siderophores. Both rats, one injected with [^68^Ga]Ga-DFO-B and [^67^Ga]Ga-DFO-E and the second with [^67^Ga]Ga-DFO-B and [^68^Ga]Ga-DFO-E, showed low radioactivity accumulation in non-infected organs (blood, spleen, pancreas, liver, heart and muscle) and primary excretion via kidneys. The highest uptake was observed in the infected lung tissue. In both rats, [^67^Ga]Ga-DFO-E and [^68^Ga]Ga-DFO-E revealed only slightly higher uptake in the infected lungs as compared with [^68^Ga]Ga-DFO-B and [^67^Ga]Ga-DFO-B (Figure 3). The difference in lung radioactivity accumulation between the two rats was most likely caused by a difference in the fungal load, which is responsible for the retention of radioactivity in the lung. It is noteworthy that the injection of [^68^Ga]Ga-DFO-B [21] or [^68^Ga]Ga-DFO-E [11] has previously been shown to cause negligible accumulation in the lungs of non-infected rats. In these experiments, the ratio of ^68^Ga activity in lung vs. blood was been 0.84 and 0.79 for [^68^Ga]Ga-DFO-B and [^68^Ga]Ga-DFO-E, respectively. In infected animals (in the current study), the ratio was significantly higher, with about 17 in rat 1 and 8 in rat 2 for both compounds.

### 3.3. Chemical Modification of DFO-B Improves Its Growth Promotion Activity at Acidic pH

To analyze the potential impact of pH-dependent charge on the uptake efficacy of ferrioxamine-type siderophores, we synthesized three derivatives of DFO-B with modifications of the amine-residue to neutralize the charge (DFO-B^ac^) or to cause an expected negative charge at neutral/alkaline pH (DFO-B^suc^), as described in the Material and Methods section (Figure 1).

First, the impact of the chemical modifications of DFO-B on the uptake efficacy was analyzed by testing their growth promotion activity as iron (Fe) chelates in comparison to Fe-DFO-E at different ambient pH (Figure 4). For these growth assays, we employed an *A. fumigatus* mutant strain (∆*sidA*/∆*ftrA*), which lacks both siderophore production and reductive iron assimilation [10]. This strain is unable to grow unless supplemented with high iron concentrations or a utilizable siderophore. We found that 0.1 µM Fe-DFO-E promoted the growth of ∆*sidA*/∆*ftrA* at the acidic, neutral and alkaline conditions. In contrast, 0.1 µM Fe-DFO-B promoted growth at pH 7 and 9 but not at pH 5.0, which confirms that Fe-DFO-B is a siderophore used by *A. fumigatus* and indicates utilization dependent on the ambient pH. Acetylation of the amine group of DFO-B (Fe-DFO-B^ac^), which prevents its protonation and, consequently, positive charge at acidic pH, rescued the growth promotion at pH 5. Succinylation of the terminal DFO-B amine group (Fe-DFO-B^suc^) led to a derivative that contained a carboxyl group, which was assumed to be uncharged (protonated) at the acidic pH but negatively charged (deprotonated) at the alkaline pH. In line with the negative effect of the charge, Fe-DFO-B^suc^ showed the lowest growth promotion activity at pH 9. Taken together, the fungal growth promotion activity of Fe-DFO-B was lowest at the acidic ambient pH, most likely due to its positive charge. In agreement, the derivatization that neutralized the positive charge improved the growth promotion activity at acidic conditions without influencing growth promotion efficacy at the alkaline pH.

### 3.4. Short-Term Uptake of DFO-B Displays pH Dependence

In long-term growth promotion assays, ambient pH and chemical modification might influence the compound stability. To corroborate pH- and charge-dependent uptake of DFO-B, in the next step, the impact of ambient pH and chemical modifications on short-term uptake efficacy as ^68^Ga-chelates was analyzed. Therefore, *A. fumigatus* Afs77 was grown in minimal medium, reflecting either iron sufficiency or iron limitation, to induce the expression of genes encoding SITs to activate siderophore uptake [27]. To analyze the effect of pH on siderophore uptake, the fungal cultures were adjusted before the uptake assays to ambient pH of 5, 7 and 9, respectively. In agreement with the regulation of siderophore uptake by iron and the specificity of the uptake assays, the uptake of all siderophores was significantly higher in the iron-limited cultures compared to the iron-sufficient cultures under all tested conditions (Figure 5a). Compared to [^68^Ga]Ga-DFO-E, the uptake efficacy of [^68^Ga]Ga-DFO-B was only approximately 20% at pH 5, and increased to about 50% at pH 7 and 80% at pH 9 (Figure 5b). These data underline the pH dependence of [^68^Ga]Ga-DFO-B compared to [^68^Ga]Ga-DFO-E. [^68^Ga]Ga-DFO-B^ac^ showed similar uptake efficacy to [^68^Ga]Ga-DFO-E at pH 5 and pH 7, which is in line with the protonated amino group being responsible for the decreased [^68^Ga]Ga-DFO-B uptake in the acidic and neutral conditions. In line with the growth promotion assays, [^68^Ga]Ga-DFO-B^suc^ displayed highly decreased uptake efficacy at pH 9, again underlining the negative impact of the charge on the ferrioxamine uptake efficacy.

## 4. Discussion

Imaging of fungal infections, such as aspergillosis, has been approached with several strategies over the last decades using various tracers, including labeled antifungals and peptides. A recent promising approach is the use of the *Aspergillus*-specific humanized monoclonal antibody hJF5 labelled with ^64^Cu as the PET tracer and NODAGA as the Cu chelator [28].

Here, we demonstrate that the ^68^Ga-labelled siderophore DFO-B is highly effective for the in vivo imaging of pulmonary *A. fumigatus* infections in a rat aspergillosis model. The main advantages of using [^68^Ga]Ga-DFO-B compared to non-siderophore radiotracers appear to be (i) the use of ^68^Ga as a radiolabel, enabling the on-site preparation of the radiotracer at the time of need via reliable automated radiosynthesis without the requirement of the cyclotron; (ii) the short half-life (68 min) of ^68^Ga, implying a low radiation burden on patients compared, for example, to the 12.7 h half-life of ^64^Cu [29]; (iii) the low molecular mass of [^68^Ga]Ga-DFO-B, resulting in excellent pharmacokinetics, such as the fast distribution and rapid renal excretion lacking accumulation in other organs [21]; (iv) the active uptake mechanism via SITs, leading to the accumulation of the radiotracer within the target organism; and (v) the repurposing of a well-established clinically used medication, Desferal^®^, which has a long history for use in metal chelation therapy as well as in nuclear medicine, as a bifunctional chelating agent enabling the radiolabeling of wide range of biomolecules with different radionuclides [30,31,32].

In previous studies [11], the ^68^Ga-labelled siderophores TAFC and DFO-E have been shown to mediate effective molecular imaging of pulmonary *A. fumigatus* infections in rat aspergillosis models. In contrast to DFO-B, however, these siderophores are not yet approved for clinical use or necessary GMP (good manufacturing practice)-compliant production. In addition, safety studies constitute significant hurdles for their application in humans.

Notably, the utilization of siderophores displays a certain species specificity. In vitro studies have revealed the uptake of [^68^Ga]Ga-TAFC by *A. fumigatus*, *Rhizopus arrhizus* and *Fusarium solani*, but no significant uptake by *Aspergillus terreus, Aspergillus flavus* or *Candida albicans* or the bacterial species *Pseudomonas aeruginosa*, *Klebsiella pneumoniae*, *Staphylococcus aureus* and *Mycobacterium smegmatis* [33]. In comparison, [^68^Ga]Ga-DFO-E displayed the highest uptake by *A. fumigatus* and considerable uptake by *A. terreus*, *A. flavus*, *R. oryzae* and *F. solani*, as well as the bacterial species *S. aureus* [33]. [^68^Ga]Ga-DFO-B uptake was found in bacterial species such as *S. aureus, P. aeruginosa* and *Streptococcus agalactiae*, as well as in the fungal species *A. fumigatus*, while *Escherichia coli* and *K. pneumoniae* appeared to lack significant uptake [12,21]. Taken together, DFO-E and DFO-B have a broad spectrum covering bacterial and fungal species, while TAFC appears to be specific to fungal TAFC utilizers. Whether this is advantageous and how this specificity impacts the clinical application remains to be established by clinical studies showing the principle potential of imaging infections in humans with [^68^Ga]Ga-DFO-B by means of PET. Regarding fungal infections, the uptake of [^68^Ga]Ga-DFO-B by certain Mucorales might be an opportunity to diagnose these otherwise difficult-to-characterize infections. Moreover, localization of the infection and therapy monitoring might be important applications of this imaging methodology.

Previous data indicated significantly lower in vitro uptake of DFO-B compared to the TAFC and DFO-E metal complexes in *A. fumigatus* [12]. In contrast to DFO-E and TAFC, which are uncharged cyclic siderophores, DFO-B is a linear molecule with an amine group that is positively charged at neutral or acidic pH (Figure 1). As shown here by growth assays and short-term uptake assays, DFO-B uptake efficacy was low at acidic pH and increased at neutral and alkaline pH. The chemical acetylation of the amine group leading to the uncharged DFO-B^ac^ improved the utilization of the iron complex as the iron source in the long-term growth assays, as well as the short-term uptake of the ^68^Ga-complex at acidic and neutral pH. These data indicate that the charge of DFO-B likely decreased the uptake efficacy. In line, we found that uptake of DFO-B^suc^, which is assumed to have a negative charge at basic pH, was inefficiently taken up at pH 9.

These data also elucidate the reason for the poor [^68^Ga]Ga-DFO-B uptake by *A. fumigatus* observed in a previous study [11], as the pH of *A. fumigatus* cultures grown under iron starvation conditions is usually acidic due to the fungal acidification of the medium [34]. Efficient uptake of [^68^Ga]Ga-DFO-B by *A. fumigatus* in a rat infection model can be explained by the approximately neutral pH in the mammalian system [35]. In line with these results, reasonable uptake of [^68^Ga]Ga-DFO-B was demonstrated here in vitro at a neutral pH. Moreover, we show here that the in vivo uptake efficacy of [^67/68^Ga]Ga-DFO-B and [^67/68^Ga]Ga-DFO-E by *A. fumigatus* is similar in a rat infection model.

During infection, pathogens usually encounter iron limitation due to nutritional immunity [36,37], which leads to the activation of siderophore uptake [8]. However, the uptake efficacy not only depends on the amount and activity of available SITs, but also on transporter/siderophore cooperation. Here, we demonstrated that the charge of a siderophore has a significant impact on the uptake efficacy. As the efficient uptake of a radiolabeled siderophore by a pathogen is crucial for effective imaging, these findings might help to improve the properties of siderophores used for imaging. In a proof-of-concept study, Pfister et al. previously demonstrated that certain siderophores could be used in a theranostic approach, in which antimicrobial therapy is combined with a diagnostic approach [38]. In this study, ^68^Ga-imaging was combined with the fusion of antimicrobials to the siderophore to exploit the so-called Trojan horse approach. The present study demonstrates that the charge of a siderophore has a significant impact on its uptake and that this must be considered when modifying siderophores for diagnostic and/or therapeutic purposes.

## 5. Conclusions

In this study, we demonstrated, for the first time, the successful application of [^68^Ga]Ga-DFO-B for imaging of *A. fumigatus* infections. Growth and short-term uptake assays with *A. fumigatus* demonstrated that Fe-DFO-B and [^68^Ga]Ga-DFO-B were taken up by the fungus in a pH-dependent manner. DFO-B has a long-standing history in clinics, and licensed pharmaceutical-grade ^68^Ge/^68^Ga-generators are already used for imaging, which might facilitate the approval of [^68^Ga]Ga-DFO-B as an imaging tool for *A. fumigatus* infections. As ferrioxamines are also utilized by several other fungi, the use of [^68^Ga]Ga-DFO-B-mediated imaging may allow different clinical applications. However, this must be verified in controlled clinical trials.

## Figures and Tables

**Figure 1 jof-07-00734-f001:**
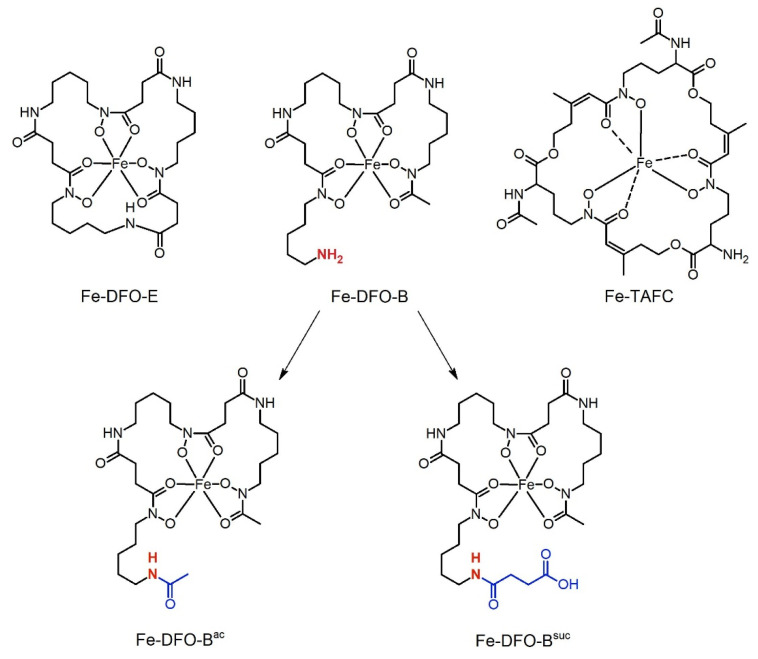
Chemical structures of ferric DFO-E, DFO-B and TAFC, as well as chemically modified DFO-B derivatives DFO-B^ac^ and DFO-B^suc^. Chemical modifications of the amino group (in red) are shown in blue.

**Figure 2 jof-07-00734-f002:**
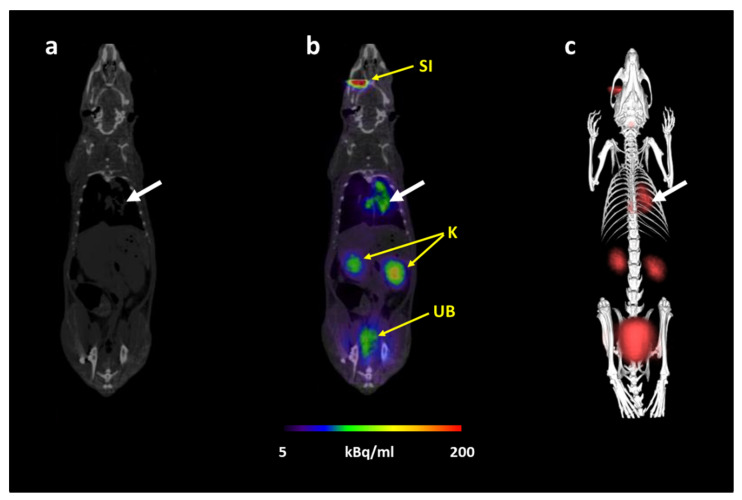
Static PET/CT images of [^68^Ga]Ga-DFO-B in pulmonary *A. fumigatus*-infected Lewis rats 45 min after injection: (**a**) CT coronal slice, (**b**) fused PET/CT coronal slice and (**c**) 3D volume rendered PET/CT image. White arrow indicates *A. fumigatus* infection, SI = site of injection, K = kidneys, UB = urinary bladder.

**Figure 3 jof-07-00734-f003:**
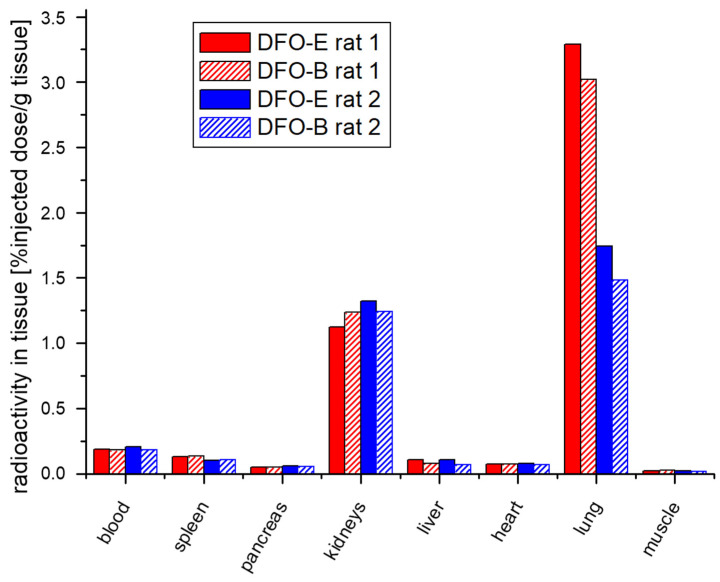
Comparison of the ex vivo biodistribution of radiolabeled DFO-B and DFO-E, respectively, expressed as % injected dose per organ.

**Figure 4 jof-07-00734-f004:**
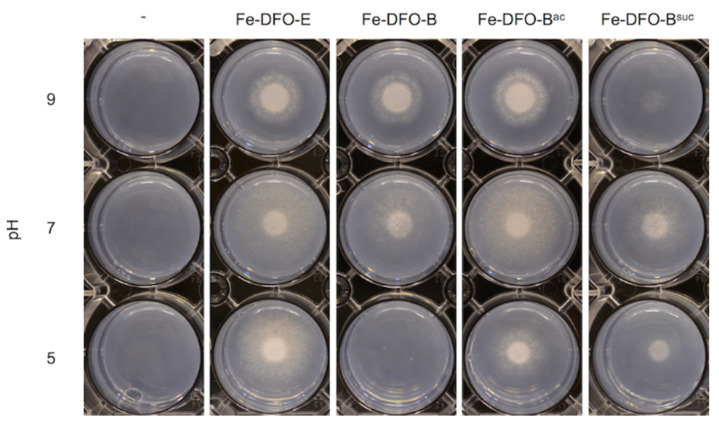
The growth promotion activity of Fe-DFO-B depends on the ambient pH and can be modulated by chemical derivatization. In total, 10^4^ spores of *A. fumigatus* strain *∆sidA/∆ftrA* were point-inoculated on the solid minimal medium with the indicated pH, lacking (-) or containing 0.1 µM of the indicated iron-loaded siderophore. Plates were incubated for 48 h at 37 °C.

**Figure 5 jof-07-00734-f005:**
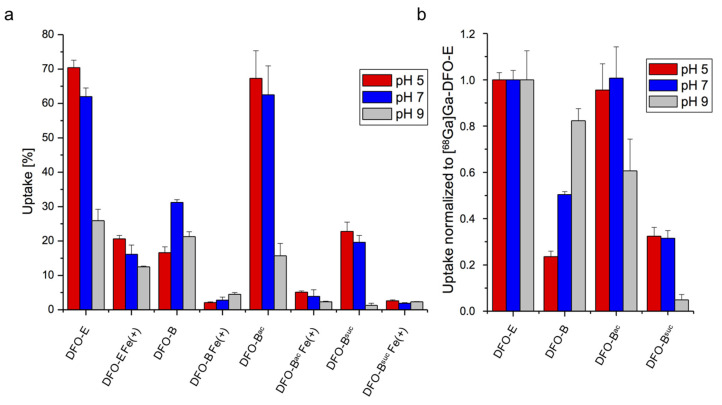
Short-term uptake efficacy of radiolabeled DFO-B depends on the ambient pH. Uptake of ^68^Ga complexed by DFO-B and derivatives under iron limitation and sufficiency (Fe(+)) is shown (**a**) normalized to total siderophore activity and (**b**) normalized to [^68^Ga]Ga-DFO-E uptake.

## Data Availability

Data is contained within the article.

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
