# Peer review of "Desferrioxamine B-Mediated Pre-Clinical In Vivo Imaging of Infection by the Mold Fungus Aspergillus fumigatus"

_jof, 2021, doi:10.3390/jof7090734_

Round 1
Reviewer 1 Report
- While A. fumigatus is the most commonly isolated Aspergillus spp., depending on the region (geographical) and anatomy (i.e. ear) A. flavus, A. terreus or A. niger complex or more common. Please add to your discussion the use of DFO-B of these other Aspergillus species., as well as the other common mucorales. While Aspergillus diagnostics is challenging, mucorales diagnostics is even more difficult (diffucult to culture, no serum markers).
- How were these rats kept? in controlled environment or area with outdoor air? humans inhale a lot of Aspergillus spores. Do you have imaging of rats breathing outside air without aspergillus inoculation as control?
- The aspfum Afs77 strain and mutant strain sidA/ftrA were used. How were these strains acquired and how was species identification verified? Please add to methods.
- Results section contains a lot of other references and discussion. I would like to see only your results in the result section.
- Humans with chronic pulmonary aspergillosis, COPD and other chronic lung diseases have colonisation and co-infections with many bacteria. S. pneumoniae, Haemophilus spp., Moraxella catarrhalis, TB/non-tuberculous mycobacteria, how do these take up DFO-B?
- Would like to see a (theoretical) discussion on sensitivity/specificity/NPV versus current used imaging modalities. Where would this modality fit, what unmet needs.
Author Response
Dear Reviewer 1,
Thank you very much for handling our paper! We appreciate the constructive comments and suggestions and amended the manuscript accordingly. We addressed all points and our specific responses are given below (in blue and italics). The original manuscript was revised (the changes have been marked in yellow in the doc file). We think that the manuscript has been significantly improved and we hope that it now meets the criteria for publication in JoF.
Thank you again for your efforts!
Hubertus
________________________________
Univ.-Prof.Mag.Dr. Hubertus Haas
Director, Institute of Molecular Biology
Innsbruck Medical University
Innrain 80-82
A-6020 Innsbruck, Austria
Phone 0043-512-9003-70205
FAX: 0043-512-9003-73100
Email: hubertus.haas@i-med.ac.at
http://mol-biol.i-med.ac.at/
__________________________________________________________________________________________
Reviewer 1
- While A. fumigatus is the most commonly isolated Aspergillus spp., depending on the region (geographical) and anatomy (i.e. ear) A. flavus, A. terreus or A. niger complex or more common. Please add to your discussion the use of DFO-B of these other Aspergillus species., as well as the other common mucorales. While Aspergillus diagnostics is challenging, mucorales diagnostics is even more difficult (difficult to culture, no serum markers).
As we agree that this is certainly an important issue, we extended the discussion in this regard without extensively lengthening it. At the end clinical trials have to show the exact imaging indications for such a methodology. Here, we present data proving that the methodology principally works for A. fumigatus and show that one has to consider pH-dependent efficacy of DFO-B uptake, when comparing in vitro and in vivo studies.
How were these rats kept? in controlled environment or area with outdoor air? humans inhale a lot of Aspergillus spores. Do you have imaging of rats breathing outside air without aspergillus inoculation as control?
Rats were kept in a controlled environment (“animal house”) with appropriate filters. We therefore only have control images of non-infected animals under these controlled conditions. It would be practically not be allowed to keep animals outside such controlled environments for animal experimentation. However, we believe that keeping rats in normal or even highly spore-containing environment would not cause spore visualization by PET (unless they cause an infection), as the signal requires growth (metabolic activity) of the fungus in combination with iron starvation – spores do not take up siderophores. For illustration: the aspergillosis model works only in immunosuppressed animals and during establishment of the aspergillosis model in our first approaches, we had a number of animals which did not develop aspergillosis due to insufficient immunosuppression: those rats were also negative in 68Ga-siderophore-mediated PET imaging.
- The aspfum Afs77 strain and mutant strain sidA/ftrA were used. How were these strains acquired and how was species identification verified? Please add to methods.
We included a reference for A. fumigatus strain AfS77 (PMID: 20656854). This is our commonly used ”lab strain”, which is continuously verified by various phenotyping assays. The origin of A. fumigatus mutant strain ∆sidA∆ftrA, which was generated in our lab in a previous study, was already referenced in Material and Methods ([10]).
- Results section contains a lot of other references and discussion. I would like to see only your results in the result section.
We restructured the text and largely eliminated references and discussion in the Results section.
- Humans with chronic pulmonary aspergillosis, COPD and other chronic lung diseases have colonisation and co-infections with many bacteria. S. pneumoniae, Haemophilus spp., Moraxella catarrhalis, TB/non-tuberculous mycobacteria, how do these take up DFO-B?
DFO-B is also utilized by some bacteria such as Staphylococcus spp and P. aeruginosa, which is mentioned in the paper. Therefore, the specificity of PET imaging with 68Ga-DFO is certainly a point of discussion. We have extended the discussion session to clarify this.
- Would like to see a (theoretical) discussion on sensitivity/specificity/NPV versus current used imaging modalities. Where would this modality fit, what unmet needs.
Our study represents a proof of principle of the suitability of [68Ga]Ga-DFO-B for preclincical imaging of infection with A. fumigatus. The clinical applications and modalities of this methodology have, however, to await controlled clinical trials. We extended the discussion to clarify this.
Reviewer 2 Report
Dear authors,
I reviewed the paper “Desferrioxamine B-mediated pre-clinical in vivo imaging of infection by the mold fungus Aspergillus fumigatus”. Please find my comments below.
Introduction
Line 60. Maybe it could be interesting for the reader to cite also the more common name of desferrioxamine B (= deferoxamine).
Line 86-89. This is an outlook of the results and should be avoided in the introduction. At this point of the introduction, the aim of the study should be explained. For example: The aim of the present study was to evaluate [68Ga]Ga-DFO-B for in vivo imaging of aspergillosis in a rat infection model, and ...
Materials and Methods
Line 99. Please cite the reference of the endoscope used.
Line 159. Please define DIPEA.
Line 178: Please define RT.
Results
In parts of this section the results of the experiments are mixed up with discussion. The results section should contain only unjudged results. Please move the discussion of the results to the discussion section of the paper.
Discussion
Line 336. Rhizopus oryzae has been renamed to Rhizopus arrhizus.
Line 335-346. What is not clear to me is if the authors have taken into account that [68Ga]Ga-DFO-B most likely is also incorporated by Rhizopus arrhizus (as already shown for [68Ga]Ga-TAFC and [68Ga]Ga-DFO-E), the most common pathogen in mucormycosis (and mucormycosis is in Europe the 2nd most common filamentous fungal infection after aspergillosis). A well-known and documented risk factor for mucormycosis is deferoxamine therapy (for example: Roden et al, 2005, Clin Infect Dis). For me as a clinician it is important to know if the pulmonary infiltrates of a high-risk patients with suspected filamentous fungal infection belong to an Aspergillus or a Mucorales infection. In Europe this has major therapeutic consequences as first-line for aspergillosis is voriconazole or isavuconazole and first-line for mucormycosis is liposomal amphotericin B (isavuconazole only when AMB is not suitable). For aspergillosis voriconazole is superior than AMB, but Mucorales are voriconazole resistant. So what do I do? Which drug should I give to the patient? I don’t see how [68Ga]Ga-DFO-B could improve this problem! For me this problem limits the usability of the technique.
Conclusions
Line 387-388: Why should it be an advantage that [68Ga]Ga-DFO-B is also utilized by several other fungi?
Author Response
Dear Reviewer 2,
Thank you very much for handling our paper! We appreciate the constructive comments and suggestions and amended the manuscript accordingly. We addressed all points and our specific responses are given below (in blue and italics). The original manuscript was revised (the changes have been marked in yellow in the doc file). We think that the manuscript has been significantly improved and we hope that it now meets the criteria for publication in JoF.
Thank you again for your efforts!
Hubertus
________________________________
Univ.-Prof.Mag.Dr. Hubertus Haas
Director, Institute of Molecular Biology
Innsbruck Medical University
Innrain 80-82
A-6020 Innsbruck, Austria
Phone 0043-512-9003-70205
FAX: 0043-512-9003-73100
Email: hubertus.haas@i-med.ac.at
http://mol-biol.i-med.ac.at/
__________________________________________________________________________________________
Reviewer 2
Introduction
Line 60. Maybe it could be interesting for the reader to cite also the more common name of desferrioxamine B (= deferoxamine).
Included.
Line 86-89. This is an outlook of the results and should be avoided in the introduction. At this point of the introduction, the aim of the study should be explained. For example: The aim of the present study was to evaluate [68Ga]Ga-DFO-B for in vivo imaging of aspergillosis in a rat infection model, and ...
As suggested we redplaced the outlook of the results by the aim of the study.
Materials and Methods
Line 99. Please cite the reference of the endoscope used.
The ”Karl Storz TELE PACK VET X LED system” includes the endoscope. We are sorry for misspelling the company (”Storz” intead of ”Stroz”). We corrected the mistake and included more information by providing a website link.
Line 159. Please define DIPEA.
DIPEA was defined (N,N-Diisopropylethylamine)
Line 178: Please define RT.
”RT” was replaced by room ”temperature”.
Results
In parts of this section the results of the experiments are mixed up with discussion. The results section should contain only unjudged results. Please move the discussion of the results to the discussion section of the paper.
We largely eliminated discussion of data in the Results section.
Discussion
Line 336. Rhizopus oryzae has been renamed to Rhizopus arrhizus.
Corrected
Line 335-346. What is not clear to me is if the authors have taken into account that [68Ga]Ga-DFO-B most likely is also incorporated by Rhizopus arrhizus (as already shown for [68Ga]Ga-TAFC and [68Ga]Ga-DFO-E), the most common pathogen in mucormycosis (and mucormycosis is in Europe the 2nd most common filamentous fungal infection after aspergillosis). A well-known and documented risk factor for mucormycosis is deferoxamine therapy (for example: Roden et al, 2005, Clin Infect Dis). For me as a clinician it is important to know if the pulmonary infiltrates of a high-risk patients with suspected filamentous fungal infection belong to an Aspergillus or a Mucorales infection. In Europe this has major therapeutic consequences as first-line for aspergillosis is voriconazole or isavuconazole and first-line for mucormycosis is liposomal amphotericin B (isavuconazole only when AMB is not suitable). For aspergillosis voriconazole is superior than AMB, but Mucorales are voriconazole resistant. So what do I do? Which drug should I give to the patient? I don’t see how [68Ga]Ga-DFO-B could improve this problem! For me this problem limits the usability of the technique.
We agree that the specificity is a certain limitation. However, the use of different siderophores might improve the specificity issue in the future. Moreover, at this stage development is still far from the clinical application. Furthermore, it should be considered that besides the mere diagnosis of infection also the exact localization of infection and therapy monitoring to control efficacy of administered antifungal treatment might be important applications. Of course, this has to be clarified in controlled clinical trials. We extended the discussion in this regard.
Conclusions
Line 387-388: Why should it be an advantage that [68Ga]Ga-DFO-B is also utilized by several other fungi?
We agree that this sentence at the end might be misleading and have reworded it.
Round 2
Reviewer 1 Report
Accept in present form.
Reviewer 2 Report
Dear authors,
Thank you for the revised and improved version of the manuscript. I have no further comments.
Best regards